# Dietary Effects of Different Proportions of Fermented Straw as a Corn Replacement on the Growth Performance and Intestinal Health of Finishing Pigs

**DOI:** 10.3390/ani15030459

**Published:** 2025-02-06

**Authors:** Xiaoguang Ji, Wenfei Tong, Xiangxue Sun, Lei Xiao, Mengjun Wu, Peng Li, Yonggang Hu, Yunxiang Liang

**Affiliations:** 1National Key Laboratory of Agricultural Microbiology, College of Life Science and Technology, Huazhong Agricultural University, Wuhan 430070, China; xiaoguang_120@126.com; 2Hubei Key Laboratory of Animal Nutrition and Feed Science, Wuhan Polytechnic University, Wuhan 430023, China; tong1wenfei@163.com (W.T.); wumengjun93@163.com (M.W.); lp1536698031@163.com (P.L.); 3Hubei Lan Good Microbial Technology Co., Ltd., Yichang 443100, China; xiangxue1108@gmail.com (X.S.); xiaol@lgzwbio.com (L.X.)

**Keywords:** fermented straw, finishing pigs, growth performance, intestinal health

## Abstract

Uncovering the potential value of corn straw is one of the strategies used to develop monogastric animal feed resources. The aim of the present study was to explore the dietary effects of different proportions of fermented straw as a replacement for corn on the growth performance and intestinal health of finishing pigs. Fermented feed supplementation was associated with the maintenance of immune functionality, antioxidant capacity, and intestinal integrity. Replacing corn with a total of 5% fermented straw in the diet improved the lipid metabolism and intestinal microbiota, while 10% fermented straw impaired growth performance. Therefore, replacing corn with 5% fermented straw in the diets of finishing pigs may be a practical and economical strategy for improving their intestinal health without detrimental effects.

## 1. Introduction

The food and feed demand in China is projected to keep increasing in the coming decades, and a recent analysis predicted that the demand for feed grain on the Chinese mainland will reach at least 389.6 million tons in 2030 [1]. A lack of feed has become the main bottleneck restricting the development of animal husbandry. With the serious competition for food between humans and animals, developing animal feed resources and sustainable livestock production is imperative [2].

The world produces about 7 billion tons of straw annually, of which China produces about 900 million [3]. Rice straw, corn straw, and wheat straw are rich in lignin, which is considered one of the most renewable resources in the world. However, most of the straw is either used as fertilizer, abandoned, or simply burned, which results in a great waste of resources and significant environmental pollution [4]. Fully exploring and utilizing the potential value of crop straw by using biotechnology is one of the most effective means to alleviate the global shortage of food resources [5].

Crop straw is usually specially treated with biophysical or chemical methods, such as sodium hydroxide treatment or the addition of urea, in order to improve digestibility and nutritional value [6,7]. Studies have found that millet straw combined with corn straw can improve the blood biochemical metabolism of fattening lambs [8]. The mushrooms *P. eryngii* and *P. sajor-caju* improve the nutritional value of corn straw as ruminant feed [9]. Adding a concentrated mixture of corn malt–acrylic acid to beef cattle diets can improve the digestibility and growth performance of beef cattle [10]. Feed fiber composition is usually considered to be one of the factors that has an impact on the digestive tract, leading to changes in the composition of the microbiota [11]. Probiotics play an essential role in intestinal health, and appropriate intestinal microbiota regulation may positively affect digestion, immune responses, and the absorption of nutrients [12]. Thus, improving the microbiota structure is one of the primary goals in the livestock and poultry industries since it results in health benefits for hosts and subsequently increases growth performance, which provides more choices for the promotion of strong and efficient productivity. Currently, feed supplemented with fermented corn straw is widely used for ruminants but is rarely used for pigs or other monogastric animals. Utilizing an appropriate proportion of fermented straw to replace a proportion of corn in pig feed could help save costs. In addition, 10% fermented, extruded corn straw can increase the number of litters per pregnant sow [13], and fermented straw products after ball milling can significantly improve pig growth, along with the digestibility and nutritional value of feed [14]. However, the effects of replacing corn with different proportions of fermented straw in feed on the growth performance and the intestinal health of finishing pigs are still unclear.

Our original research involved the development of a CO_2_ neutralization procedure during the thermophilic anaerobic digestion of lime-treated corn straw, which is recommended as a more efficient and affordable strategy for fermentation [15]. Based on the previous results, the present study aimed to further explore the effects on growth performance and intestinal health when replacing corn with different proportions of fermented straw in the diet of finishing pigs. 

## 2. Materials and Methods

### 2.1. Experiment Design and Animal Management

A single-factor design was employed to study the effects of replacing corn with different proportions of fermented straw in the diet of finishing pigs on growth performance and intestinal health. A total of 275 commercial healthy finishing pigs aged 126 days (average body weight, 82.96 ± 3.07 kg) were randomly allocated into three groups: the control group (CTR, basal diet), the 5% fermented straw group (FJJG5, replacing 5% of the corn), and the 10% fermented straw group (FJJG10, replacing 10% of the corn). Six replicates were used in each group, with 14–16 pigs per replicate. The diet was provided by Liuhe Tianen Feed Co., Ltd. (Dezhou, China). The basal diet was formulated according to the NRC 2012 nutritional requirements standard for piglets, and the formula and nutritional composition are shown in Table 1. The pigs were fed twice daily in closed, semi-slatted, concrete-floor piggeries, with free access to food and water. Body weight and food intake were recorded every two weeks until the pigs from the control group reached the commercial finished weight (130 kg). Then, the average daily gain (ADG), average daily feed intake (ADFI), and the feed-to-gain ratio (F/G) were calculated. On day 39 of the experiment, one animal from each replicate was slaughtered for sampling and further analysis. Blood, intestinal tissue, and colonic content samples were collected and subjected to biochemical analysis or stored at −80 °C until further analysis. The experimental protocol and the management and care of the pigs were approved by the Animal Care and Use Committee of New Hope Liuhe Co., Ltd. (Chengdu, China). The code for ethical inspection was IAS 2023-25.

### 2.2. Sample Collection

Blood samples were collected from the jugular veins of piglets on D39, and then the animals were slaughtered under sodium pentobarbital anesthesia (50 mg/kg body weight, BW). The abdomen was opened immediately, and the intestine was separated from the mesentery onto a chilled stainless-steel tray. Then, 5 cm segment samples were cut at the mid-jejunum and mid-colon, and the samples were gently flushed with ice-cold PBS [16]. In addition, the colonic content was transferred to an Eppendorf tube and then rapidly frozen in liquid nitrogen and stored at −80 °C until analysis.

### 2.3. Blood Sample Collection and Blood Biochemical Measurements in Plasma

Blood samples were collected from the anterior vena cava into either heparinized or serum vacuum tubes (Becton-Dickinson Vacutainer System, Franklin Lake, NJ, USA). The blood samples were centrifuged at 3000 rpm for 10 min at 4 °C to obtain plasma; then, both the plasma and serum were stored at −80 °C until analysis. The concentrations of biochemical parameters in plasma were measured with corresponding kits using a Hitachi 7060 Automatic Biochemical Analyzer (Hitachi, Japan) [17].

### 2.4. Intestinal Morphology

The intestinal histomorphology was assessed following a previous methodology [18]. Small intestine samples (1 cm long) were fixed in 4% paraformaldehyde. Then, the fixed samples were dehydrated and embedded in paraffin, sectioned at a thickness of 4 mm, and stained with hematoxylin and eosin. A morphological examination was conducted with a light microscope (Leica microsystems, Wetzlar, Germany) with Leica Application Suite image analysis software (Leica microsystems, Wetzlar, Germany). Several intact, well-oriented crypt–villus units were randomly selected and measured in triplicate per section. The intestinal villus height (VH) and crypt depth (CD) were measured to calculate the ratio of villus height to crypt depth (VH/CD).

### 2.5. Antioxidant Capacity and Oxidation-Relevant Products in Serum and Intestine

Serum, jejunum, and colon samples were used for the analysis of antioxidative enzymes and related products. The total superoxide dismutase (T-SOD) activity, total antioxidant capacity (T-AOC), and malondialdehyde (MDA) content were determined using commercially available kits (Nanjing Jiancheng Bioengineering Institute, Nanjing, China). Assays were performed in triplicate according to the instructions of a previous publication [19].

### 2.6. Determination of Intestinal Digestive Enzyme Activity and IgA Concentration

The intestinal disaccharidases and lipase activities of the colonic content and IgA concentrations of the jejunum and colon were determined using commercially available kits (Nanjing Jiancheng Bioengineering Institute, Nanjing, China). Assays were performed in triplicate, following the manufacturer’s instructions.

### 2.7. Detection of mRNA Expression Levels via Real-Time Quantitative PCR

Total RNA from the jejunum and colon was extracted using an RNAiso Plus (Takara, Dalian, China) reagent. Then, cDNA was synthesized using a PrimeScript^®^RT reagent kit with gDNA Eraser (Takara, Dalian, China). A real-time quantitative PCR was performed using the SYBR^®^ Premix Ex Taq TM (Takara, Dalian, China) on an Applied Biosystems 7500 Fast RT-qPCR System (Foster City, CA, USA) under the following conditions: 95 °C, 30 s; 95 °C, 5 s; 60 °C, 34 s, a total of 40 cycles; 95 °C, 15 s; 60 °C, 1 min; and 95 °C, 15 s. The ribosomal protein L4 (RPL4) gene was employed as a reference in the present study, and the relative expression of genes was calculated using the 2^−ΔΔCt^ technique reported previously [20]. The primer sequences used for this study are listed in Table 2.

### 2.8. Colon Microbiota Analysis

Genomic DNA was extracted using a DNA extraction kit. The purity and concentration of the DNA were then tested using 1% agarose gel electrophoresis, and the sample was diluted to 1 ng/μL using sterile water and used as a template. The universal primers 515F and 806R of the 16S rDNA gene V4 region were used to identify bacterial diversity according to previously described methods [21]. After amplification, PCR products were run on a 2% agarose gel and were purified using a QIAquick Gel Extraction Kit (Qiagen, Hilden, Germany). Purified amplicons were pooled in equimolar amounts, and their paired-end reads were sequenced on an Illumina HiSeq2500 PE250 platform (Illumina, San Diego, CA, USA) at Novogene Bioinformatics Technology Co., Ltd. (Beijing, China). Sequence processing and bioinformatics analysis were conducted in accordance with a previous study [22]. In brief, raw tags were generated by merging paired-end reads using FLASH software (v1.2.11). High-quality clean tags were obtained via QIIME analysis, and chimera sequences were removed to obtain effective tags using the UCHIME algorithm. Sequences were analyzed with UPARSE software and clustered into operational taxonomic units (OTUs) at a similarity level of 97%. Each OTU was annotated with the Greengenes database. Beta diversity was evaluated by using principal coordinate analysis (PCOA) to show the differences of bacterial community structures, and the significance of separation was tested via ANOSIM. LEfSe software (Version 1.0) was used to obtain the LEfSe results, and the LDA scores were greater than 3.2. PICRUSt analysis and R (Version 2.15.3) were used to predict the functional potential of bacteria communities. OTUs were normalized by copy number, and metagenome prediction was further categorized into the Kyoto Encyclopedia of Genes and Genomes (KEGG) at level 3. The sequencing data are openly available in the NCBI database with the entry number PRJNA1180796.

### 2.9. Statistical Analysis

All data were analyzed via a one-way ANOVA procedure in SPSS 23.0 software (SPSS Inc., Chicago, IL, USA) and are expressed as the mean ± SD. Duncan’s test was used for multiple comparisons. A value of *p* < 0.05 was taken to indicate statistical significance.

## 3. Results

### 3.1. Growth Performance

As shown in Table 3, fermented straw did not influence the body weight of the finishing pigs from D0 to D39. During the initial stage (D0–D14), the finishing pigs in the FJJG5 group exhibited a significantly increased ADG compared with the FJJ10 group (*p* < 0.05). The finishing pigs in both the FJJG5 and FJJG10 groups had an increased F/G ratio during the second stage (D14–D28) compared with the CTR group (*p* < 0.05). During the full trial (D0–D39), the pigs in the FJJG10 group showed a significantly decreased ADG and an increased F/G ratio (*p* < 0.05), while there was no difference between the FJJG5 group and the CTR group (*p* > 0.05). In addition, different proportions of fermented straw as a replacement for corn in the diet did not alter the mortality rate of the finishing pigs (*p* > 0.05).

### 3.2. Plasma Biochemical Parameters

Data on plasma biochemical parameters are summarized in Table 4. Compared with the control group, the FJJG5 group exhibited decreased contents of total cholesterol, phosphorus, creatinine, high-density lipoprotein, and lactate dehydrogenase (*p* < 0.05), while the FJJG10 group showed decreased activity of creatine kinase and lactate dehydrogenase (*p* < 0.05).

### 3.3. Intestinal Morphology

Data on the small intestinal morphology are summarized in Table 5. Compared with the CTR group, the FJJG10 group displayed a decreased crypt depth and, thus, an increased villus height to crypt depth ratio in the jejunum (*p* < 0.05). In addition, different proportions of fermented straw as a replacement for corn in the diet did not alter the crypt depth in the colon (*p* > 0.05).

### 3.4. Antioxidant Capacity

The results are shown in Table 6. Compared with the CTR group, the FJJG5 and FJJG10 groups demonstrated significantly increased T-SOD activity and MDA content in the colon (*p* < 0.05). In addition, the FJJG10 group exhibited significantly decreased MDA content and T-AOC activity in the serum (*p* < 0.05).

### 3.5. Intestinal Digestive Enzyme Activity and IgA Concentration Antioxidant Capacity

The results are given in Table 7. The FJJG5 group exhibited increased activities of jejunal disaccharidase and lipase (*p* < 0.05), while there was no difference in IgA concentration between the groups (*p* > 0.05).

### 3.6. mRNA Expression Levels in Jejunum and Colon

The results are given in Table 8. Compared with the CTR group, the FJJG5 group showed an increased relative expression of *APOA4*, *LPL*, and *MUC2* but decreased *SLC7A7* and *IL-10* in the jejunum and *APOA4* in the colon (*p* < 0.05). The FJJG10 group presented a decreased relative expression of *SLC7A7* and *IL-10* in the jejunum and decreased *MMP13*, *KCNJ13*, *APOA4*, *SLC7A7*, *LPL*, and *IL-10* in the colon (*p* < 0.05).

### 3.7. Colon Microbiota Analysis

The results showed that the CTR and FJJG5 groups could be separated well (Figure 1A,B), proving that the colon microbiota composition of pigs fed with 5% fermented straw was significantly different. In addition, the FJJG5 group demonstrated an increased relative abundance of Lactobacillus in colon contents, while the FJJG10 group exhibited an inhibited relative abundance of streptococcus (Figure 1C–F). The results of LEfSe analysis showed that the dominant bacterial flora in the FJJG5 group were Clostridia-UCG-014, Veillonellaceae, and Proteobacteria (Figure 2). Furthermore, the functional prediction analysis results showed that the addition of 5% fermented straw may promote aminoacyl-tRNA synthesis, cell processes and transduction, ion-coupled transporters, and other pathways in the intestinal flora of the finishing pigs and inhibit amino acid metabolism and fat synthesis pathways (Figure 3A). The addition of 10% fermented straw promoted the metabolic pathways of vitamins and their cofactors in the colonic microorganisms of the finishing pigs but inhibited the metabolic pathways of carbohydrate metabolism, protein kinase, etc. (Figure 3B).

## 4. Discussion

The inclusion of dietary fibers in the diet may have a positive impact on the health and wellbeing of pigs [23]. The present study is the first to investigate the effects of replacing corn in the diet with different proportions of fermented straw, as a form of CO_2_ neutralization, on the growth performance and intestinal health of finishing pigs. The aim of the present study was to provide evidence that it is practicable, efficient, and affordable to use fermented corn straw to partly replace corn in the diet of finishing pigs in regions where corn is widely planted. The results indicated that diets with different proportions of fermented straw did not affect the final body weight of pigs. The 5% fermented straw replacement group demonstrated no negative impact on the ADG or the F/G ratio; however, with the increase in the straw replacement ratio, the ADG and the F/G ratio increased in the 10% fermented straw replacement group. In contrast, a previous study reported that there was no significant difference in the ADG level between the 10% additive group and the control group, while the 5% additive group exhibited an increased ADG level compared with the control group [14]. This is possibly attributed to the high content of beneficial metabolites and probiotic population, which have significant importance for intestinal health and feed absorption efficiency. The differences in the ADG and the F/G ratio of the pigs can be explained by digestive absorption, which may be affected by the different diets and which is the nutritional factor with the greatest influence on weight gain [24]. Previous studies concluded that the inclusion of commercial fiber preparations or fiber-rich byproducts in pig diets may impair the digestibility of energy and organic matter, and, consequently, the performance of the animals [25,26]; however, these studies did not investigate the effects of different proportions of fiber-rich byproducts such as fermented straw on pigs. In the present study, the 5% additive group showed increased activities of jejunal disaccharidase and lipase, which may improve digestive ability. A previous study indicated that feed containing 10% fermentation products resulted in stronger satiety [27], reduced the nutritional proportion in the feed, and ultimately affected the feeding efficiency, similar to the present study. Therefore, the feed with 5% fermented corn straw is more practical than the 10% diet and has no negative impact on the growth performance of finishing pigs.

Blood plays an essential role in transporting nutrients, regulating body fluid balance, maintaining internal environment stability, and participating in body immunity; thus, plasma parameters closely reflect animal health [28,29]. Fermented straw groups demonstrated decreased contents of total cholesterol and high-density lipoprotein, indicating that this diet can regulate the lipid metabolism of pigs. The lipoprotein *APOA4* is primarily synthesized by enterocytes of the small intestine and is involved in the metabolic procedure of lipid and glucose and anti-inflammatory response [30], while the lipoprotein lipase gene is a key enzyme for fat deposition. A previous study showed that high dietary fiber can significantly increase *LPL* mRNA expression levels in geese [31]. The present study found that the FJJG5 group presented an increased relative expression of *APOA4* and *LPL* in the jejunum, but the FJJG10 group exhibited decreased *APOA4* and *LPL* in the colon. Similarly, a recent study suggested that corn straw-saccharification fiber improved the reproductive performance of sows in late gestation and lactation via lipid metabolism [32]. In short, 5% fermented straw diets could improve the lipid metabolism of finishing pigs. In addition, the results of the lower plasma creatine kinase and lactate dehydrogenase levels indicated that the anti-stress capability of pigs in the fermented straw groups was enhanced, similar to a previous study that found that finger millet straw with corn straw in the diet decreased the plasma lactic dehydrogenase level in lambs [8].

Factors such as environmental, physiological, and pathological influences can easily lead to oxidative stress in the body, resulting in injury to tissue function [33]. Antioxidant enzymes like T-SOD and T-AOC play a crucial role in scavenging free radicals in the body. The activity of these enzymes directly indicates the body’s antioxidant capacity and helps maintain the balance of free radicals [34]. The fermented straw replacement group had significantly increased T-SOD activity and MDA content in the colon, while the 10% additive group exhibited significantly decreased MDA content and T-AOC activity in the serum. Another study found that saccharified corn straw increased the serum total antioxidant capacity activity of broilers in the later stage [35]. The redox state in the body is a dynamic and complex process; many factors like feed management, diet composition, fermentation methods, and species may affect the results. In addition, fermented straw decreased the relative expression of *IL-10*, but there was no difference in IgA concentration between the groups in the present study, similar to a previous study conducted in lambs [8]. The immune system is vital for health during fattening and not only affects the daily weight gain but also the disease incidence. In the present study, we did not find a difference in the mortality rate between the groups. Therefore, fermented straw did not have a negative influence on the immune system.

The growth potential of pigs depends on their intestinal health [36,37,38]. The villus height, crypt depth, and villus height/crypt depth ratio are directly connected to absorption function and are regarded as good indices of intestine structural health in animals [39]. The main component of mucus is mucin, of which mucin 2 is one of the most abundant mucins secreted by goblet cells [40]. It can form a protective mucus layer and dynamically interact with intestinal epithelial cells, microbiota, and host immune defense to maintain intestinal mucosal homeostasis [41]. The FJJG5 group showed an increased relative expression of *MUC2* in the jejunum. Interestingly, 10% fermented corn straw decreased the crypt depth, thus increasing the jejunum’s villus height/crypt depth ratio. Similarly, a recent study of intestinal morphometry found that the jejunal villus height and the villus height/crypt depth ratio of geese were significantly increased in the 15% fermented maize stover group. There were no significant differences in the 5% and 10% additive groups compared with the control group [42]. Supplementation with 20% fermented feed significantly increased the villus height of the duodenum, jejunum, and ileum in laying hens [43]. Thus, fermented straw may improve intestinal morphology if it is present in a high proportion in the diet. In addition, different proportions of fermented straw as a replacement for corn in the diet did not alter the crypt depth in the colon.

The intestinal microbiota plays an important role in the regulation of immune function, digestion, absorption, and intestinal morphology. The composition of the gut microbiome is influenced by a wide variety of factors, including species, genotype, age, diet, and living environment, with diet being of particular importance. Growing evidence indicates the beneficial effects of fermented feed on pigs’ health [44]. Fermented wheat bran increased microbiota diversity and the relative abundance of probiotics in Min pigs [45]. In the present study, the composition and structure of the colonic microbiota were significantly affected—a finding consistent with a prior study [46]. We found that the colonic microflora of the finishing pigs was dominated by Bacteroidetes and Firmicutes species, which constitute >80% of the overall microbial community therein. Firmicutes species are the primary fiber-degrading microbes present in many vertebrates, and these bacteria were more abundant in the 5% fermented corn straw group relative to the CTR and the 10% additive groups. Firmicutes species can degrade cellulose to produce polysaccharides that can in turn be utilized as an energy and carbon source by the host. These bacteria can also produce beneficial compounds, such as short-chain fatty acids [47,48]. At the genus level, an increased Lactobacillus abundance was observed in finishing pigs fed a diet containing fermented straw. Lactobacillus can produce lactic acid, thereby inhibiting the colonization of pathogenic microorganisms. Similar results conducted in laying hens were obtained in a previous study [43]. The fermented feed supplement improved the intestinal microecological environment by inhibiting pathogenic microorganisms and favoring beneficial microbiomes. According to LEfSe analysis, the beneficial effect in the fermented straw group is most likely because of the dominant bacterial flora Clostridia-UCG-014 in the FJJG5 group, which has been reported to be positively associated with bacterial diversity and barrier function in healthy individuals [49,50].

Overall, our results indicated that 5% fermented straw as a replacement for corn in the diet improved the lipid metabolism of finishing pigs, while 10% fermented straw impaired growth performance. Fermented feed supplementation was also associated with the maintenance of immune functionality, antioxidant capacity, and intestinal integrity. Furthermore, the colonic microecological environment was also improved after a 5% fermented straw replacement by increasing Lactobacillus abundance and Clostridia-UCG-014 abundance. Considering the relatively low cost of fermenting corn straw, the present results indicated that 5% fermented straw as a replacement for corn may be a practical and economical strategy for diets to improve intestinal health without detrimental effects on finishing pigs.

## 5. Conclusions

In conclusion, 5% fermented straw can improve the lipid metabolism and colon microbiota structure of finishing pigs, while 10% fermented straw has adverse effects on their growth performance and intestinal health.

## Figures and Tables

**Figure 1 animals-15-00459-f001:**
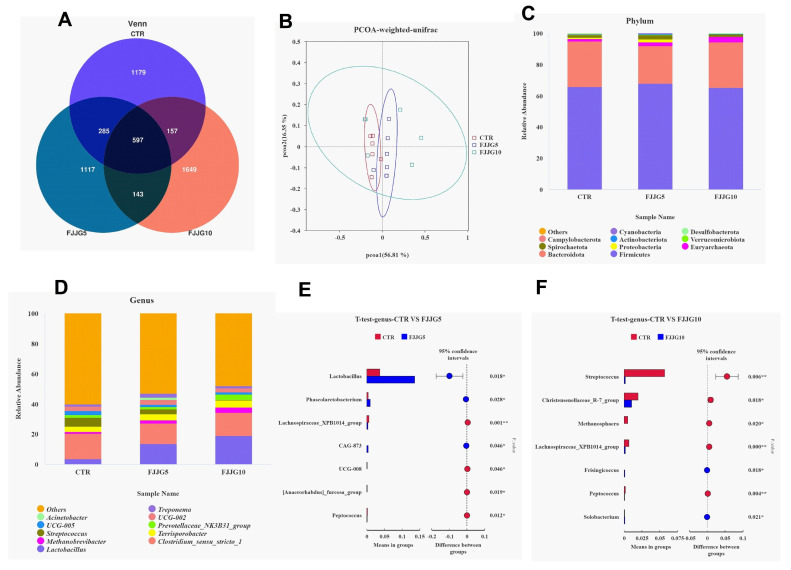
Effects of fermented straw on microbial structure of colon microbiota in finishing pigs. The Venn and PcoA diagrams are shown in (**A**,**B**), respectively. The relative abundance of bacteria on phylum and genus levels is displayed in (**C**,**D**). The results of the *t*-test are shown in (**E**,**F**). ** represents a significant difference (0.001 < *p* < 0.01), and * denotes a trend with difference (*p* < 0.05).

**Figure 2 animals-15-00459-f002:**
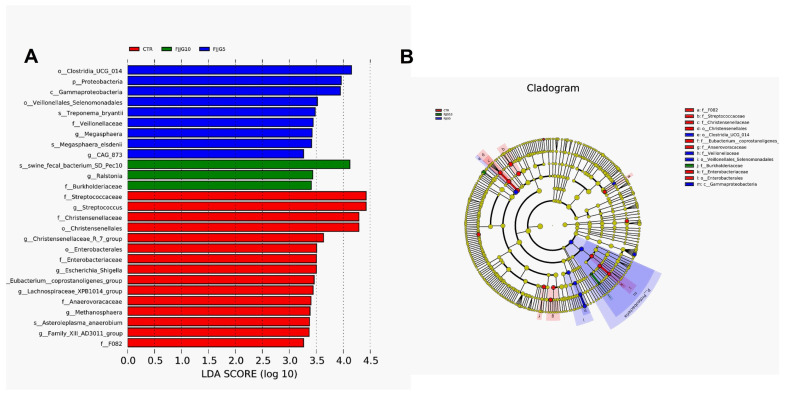
Effects of fermented straw on dominant microbiota in finishing pigs based on LEfSe analysis. (**A**), LDA analysis; (**B**), cladogram analysis.

**Figure 3 animals-15-00459-f003:**
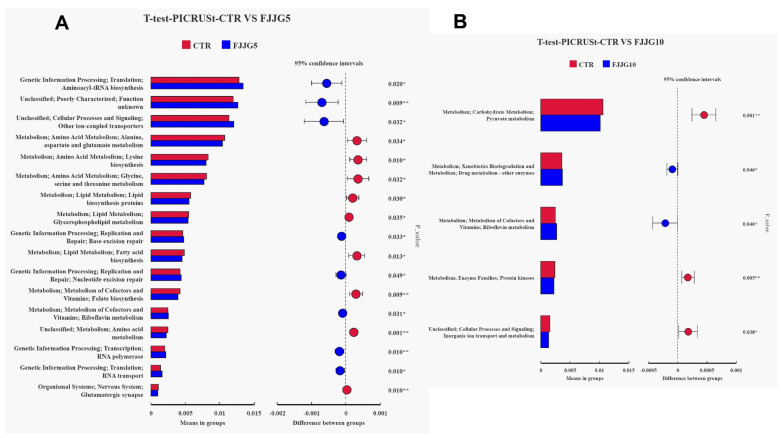
Effects of fermented straw on dominant flora and function prediction of colon microbiota in finishing pigs. The results for the CTR vs. FJJG5 group and the CTR vs. FJJG10 group are given in (**A**,**B**). ** represents a significant difference (0.001 < *p* < 0.01), and * denotes a trend with difference (*p* < 0.05).

**Table 1 animals-15-00459-t001:** The composition and nutrient levels of the basal diet and diets supplemented with fermented straw.

	CTR	FJJG5	FJJG10
Ingredient, %			
Corn	46.098	41.098	36.098
Fermented straw	0.000	5.000	10.000
Wheat	37.000	37.000	37.000
Soybean meal	8.500	8.500	8.500
Wheat bran	5.200	5.200	5.200
Limestone	1.190	1.190	1.190
L-lysine sulfate	0.700	0.700	0.700
CaHPO_4_	0.420	0.420	0.420
NaCl	0.370	0.370	0.370
Mineral premix ^1^	0.150	0.150	0.150
L-threonine	0.140	0.140	0.140
Feed enzymes	0.100	0.100	0.100
DL-Met	0.050	0.050	0.050
Microbial agents	0.030	0.030	0.030
Vitamin premix ^2^	0.022	0.022	0.022
Sodium butyrate	0.020	0.020	0.020
Cu sulfate pentahydrate	0.010	0.010	0.010
Total	100.000	100.000	100.000
Nutrient Level ^3^			
DE, Mcal/kg	3.30	3.19	3.08
NE, Mcal/kg	2.46	2.33	2.20
CP, %	13.73	13.62	13.51
Ca, %	0.62	0.62	0.62
Total P, %	0.45	0.43	0.42
Available P, %	0.25	0.24	0.24
Lysine, %	1.21	1.20	1.19
Methionine, %	0.25	0.24	0.24
Tryptophan, %	0.15	0.15	0.15
Threonine, %	0.59	0.58	0.57
Valine, %	0.58	0.57	0.55
Leucine	1.08	1.04	0.99
Isoleucine	0.48	0.47	0.46

^1^ Mineral premix contained the following substances (provided per kilogram of feed): Cu, 10 mg; Zn, 100 mg; Fe, 80 mg; Mn, 100 mg; selenium, 0.3 mg; iodine, 0.7 mg. ^2^ Vitamin premix contained the following substances (provided per kilogram of feed): vitamin A, 8000 IU; vitamin D, 3000 IU; vitamin E, 44 IU; vitamin K, 3.2 mg; vitamin B1, 3.0 mg; vitamin B2, 4.0 mg; vitamin B12, 0.025 mg; biotin, 0.0325 mg; folic acid, 2.00 mg; nicotinic acid, 15 mg; pantothenic acid, 15 mg. ^3^ Calculation level.

**Table 2 animals-15-00459-t002:** Sequences of the primers used for qPCR analysis.

Gene Name	Primer	Sequence	Amplicon Size (bp)	Accession Number
*RPL4*	Forward	5′-GAGAAACCGTCGCCGAAT-3′	146	XM_005659862.3
	Reverse	5′-GCCCACCAGGAGCAAGTT-3′		
*Mucin-2*	Forward	5′-TTCATGATGCCTGCTCTTGTG-3′	93	XM_421035
	Reverse	5′-CCTGAGCCTTGGTACATTCTTGT-3′		
*MMP13*	Forward	5′-GTGCTTCCTGATGATGATGTGC-3′	174	NM_001293090.2
	Reverse	5′-TCGCCAGAAAAACCTGTCCTT-3′		
*KCNJ13*	Forward	5′-ATGGATGTGTCGCTGGTCTTT-3′	141	XM_001926506.6
	Reverse	5′-CACAACTGCTTGCCTTTACGAG-3′		
*NHE3*	Forward	5′-AAGTACGTGAAGGCCAACATCTC-3′	341	XM_013978225.2
	Reverse	5′-TTCTCCTTGACCTTGTTCTCGTC-3′		
*APOA1*	Forward	5′-CCTTGGCTGTGCTCTTCCTC-3′	100	M_214398.1
	Reverse	5′-ACGGTGGCAAAATCCTTCAC-3′		
*APOA4*	Forward	5′-ACCCAGCAGCTCAACACTCTC-3′	122	NM_214388.1
	Reverse	5′-GAGTCCTTGGTCAGGCGTTC-3′		
*SLC7A7*	Forward	5′-TTTGGTTCCCAAGGTTGCA-3′	58	XM_013978225.2
	Reverse	5′-GCAGCTTCCTGGCATTGC-3′		
*LPL*	Forward	5′-AGCCTGAGTTGGACCCATGT-3′	128	NM_214286.1
	Reverse	5′-CTCTGTTTTCCCTTCCTCTCTCC-3′		
*IL10*	Forward	5′-CGGCGCTGTCATCAATTTCTG-3′	89	NM_214041.1
	Reverse	5′-CCCCTCTCTTGGAGCTTGCTA-3′		

**Table 3 animals-15-00459-t003:** Effects of fermented straw on growth performance of finishing pigs.

Item	CTR	FJJG5	FJJG10	*p*-Value
BW (kg)				
D0	83.11 ± 3.65	82.98 ± 2.36	82.80 ± 3.61	0.986
D14	98.95 ± 3.17	99.87 ± 2.59	97.74 ± 2.90	0.459
D28	117.81 ± 3.46	117.73 ± 3.93	115.08 ± 2.89	0.327
D39	129.45 ± 3.56	129.74 ± 3.52	126.41 ± 3.41	0.221
Initial stage (D0–D14)				
ADFI (g/d)	3011 ± 76	3109 ± 171	3002 ± 98	0.272
ADG (g/d)	1131 ± 78 ^ab^	1207 ± 95 ^a^	1067 ± 79 ^b^	0.037
F/G	2.67 ± 0.19	2.58 ± 0.14	2.83 ± 0.22	0.104
Second stage (D14–D28)				
ADFI (g/d)	3538 ± 163	3741 ± 189	3604 ± 115	0.113
ADG (g/d)	1347 ± 90	1276 ± 125	1239 ± 92	0.218
F/G	2.63 ± 0.15 ^b^	2.95 ± 0.20 ^a^	2.92 ± 0.18 ^a^	0.015
Third stage (D28–D39)				
ADFI (g/d)	3599 ± 199	3781 ± 154	3665 ± 238	0.307
ADG (g/d)	1059 ± 46	1091 ± 140	1030 ± 99	0.590
F/G	3.40 ± 0.13	3.50 ± 0.38	3.58 ± 0.30	0.577
Full stage (D0–D39)				
ADFI (g/d)	3366 ± 137	3525 ± 161	3405 ± 127	0.162
ADG (g/d)	1188 ± 37 ^a^	1199 ± 47 ^a^	1118 ± 40 ^b^	0.008
F/G	2.83 ± 0.09 ^b^	2.94 ± 0.04 ^ab^	3.05 ± 0.12 ^a^	0.003
Mortality rate (%)	2.23 ± 3.47	3.27 ± 3.58	3.2 ± 3.50	0.852

CTR, control group; FJJG5, 5% fermented straw as a replacement for corn in the diet; FJJG10, 10% fermented straw as a replacement for corn in the diet. BW, body weight; ADG, average daily gain; ADFI, average daily feed intake; F/G, feed-to-gain ratio. Duncan’s test was used for multiple comparisons of the mean. ^a,b^ Means in the same row without common superscripts differ significantly (*p* < 0.05), the same as below.

**Table 4 animals-15-00459-t004:** Effects of fermented straw on plasma biochemical parameters in finishing pigs.

Item	CTR	FJJG5	FJJG10	*p*-Value
ALB (g/dL)	45.22 ± 3.34	42.85 ± 3.47	45.74 ± 3.51	0.329
AST (U/L)	37.64 ± 10.60	30.75 ± 5.67	31.57 ± 6.20	0.279
ALT (U/L)	40.17 ± 10.42	45.00 ± 9.32	49.40 ± 5.92	0.222
ALP (U/L)	149.00 ± 45.73	157.50 ± 29.93	188.17 ± 55.91	0.315
TC (mg/dL)	3.07 ± 0.20 ^a^	2.60 ± 0.14 ^b^	2.98 ± 0.21 ^a^	0.001
TG (mg/dL)	0.70 ± 0.17	0.61 ± 0.11	0.71 ± 0.14	0.433
GLU (mg/dL)	4.95 ± 0.48	5.12 ± 0.93	5.30 ± 0.87	0.746
CA (mg/dL)	3.02 ± 0.29	2.79 ± 0.33	3.04 ± 0.20	0.265
P (mg/dL)	3.14 ± 0.40 ^a^	2.41 ± 0.45 ^b^	2.68 ± 0.45 ^ab^	0.032
CREA (mg/dL)	127.64 ± 17.67 ^a^	102.51 ± 16.58 ^b^	125.56 ± 13.64 ^a^	0.029
HDL (mg/dL)	1.39 ± 0.16 ^a^	1.19 ± 0.06 ^b^	1.40 ± 0.10 ^a^	0.010
LDL (mg/dL)	1.27 ± 0.12	1.10 ± 0.09	1.25 ± 0.14	0.056
BUN (mg/dL)	7.05 ± 1.69	6.24 ± 1.10	6.09 ± 1.13	0.426
GGT (U/L)	55.17 ± 11.11	59.80 ± 16.45	41.31 ± 8.63	0.054
CK (U/L)	1416.98 ± 445.18 ^a^	1593.01 ± 558.08 ^a^	719.96 ± 146.16 ^b^	0.006
LDH (U/L)	450.84 ± 37.00 ^a^	330.33 ± 50.53 ^b^	372.40 ± 85.00 ^b^	0.012

ALB: albumin; AST: aspartate aminotransferase; ALT: alanine amino transferase; ALP: alkaline phosphatase; TC: total cholesterol; TG: triglyceride; GLU: glucose; CA: calcium; P: phosphorus; CREA: creatinine; HDL: high-density lipoprotein; LDL: low-density lipoprotein; BUN: blood urea nitrogen; GGT: γ-glutamyltranspeptidase; CK: creatine kinase; LDH: lactate dehydrogenase. Duncan’s test was used for multiple comparisons of the mean. ^a,b^ Means in the same row without common superscripts differ significantly (*p* < 0.05).

**Table 5 animals-15-00459-t005:** Effects of fermented straw on intestinal morphology in finishing pigs.

Item		CTR	FJJG5	FJJG10	*p*-Value
Jejunum	VH	500.76 ± 59.93	469.92 ± 37.08	484.45 ± 44.03	0.551
	CD	314.34 ± 26.81 ^a^	275.56 ± 40.57 ^ab^	255.53 ± 26.57 ^b^	0.019
	VH/CD	1.64 ± 0.11 ^b^	1.77 ± 0.18 ^ab^	2.00 ± 0.27 ^a^	0.021
Colon	CD	223.84 ± 56.54	252.27 ± 93.02	258.37 ± 89.96	0.740

VH, villus height; CD, crypt depth. Duncan’s test was used for multiple comparisons of the mean. ^a,b^ Means in the same row without common superscripts differ significantly (*p* < 0.05).

**Table 6 animals-15-00459-t006:** Effects of fermented straw on antioxidant capacity in finishing pigs.

Item		CTR	FJJG5	FJJG10	*p*-Value
Serum	T-SOD, U/mL	78.22 ± 3.10	73.63 ± 9.61	72.72 ± 16.70	0.671
	MDA, nmol/mL	1.90 ± 0.18 ^a^	1.89 ± 0.32 ^a^	1.38 ± 0.25 ^b^	0.004
	T-AOC, mmol/mL	32.07 ± 4.58 ^a^	26.80 ± 5.59 ^ab^	23.08 ± 4.96 ^b^	0.025
Jejunum	T-SOD, U/mg	95.03 ± 22.52	111.83 ± 33.02	94.25 ± 20.58	0.436
	MDA, nmol/mg	1.40 ± 0.38	1.19 ± 0.32	1.24 ± 0.33	0.565
	T-AOC, mmol/mg	2.36 ± 0.92	2.33 ± 1.29	1.85 ± 1.16	0.690
Colon	T-SOD, U/mg	73.70 ± 17.90 ^b^	119.94 ± 24.29 ^a^	123.49 ± 29.61 ^a^	0.005
	MDA, nmol/mg	1.18 ± 0.24 ^b^	1.65 ± 0.25 ^a^	1.65 ± 0.45 ^a^	0.040
	T-AOC, mmol/mg	2.48 ± 0.47	2.74 ± 0.55	3.04 ± 0.31	0.135

T-SOD: total superoxide dismutase; T-AOC: total antioxidant capacity; MDA: malondialdehyde. Duncan’s test was used for multiple comparisons of the mean. ^a,b^ Means in the same row without common superscripts differ significantly (*p* < 0.05).

**Table 7 animals-15-00459-t007:** Effects of fermented straw on intestinal digestive enzyme activity and IgA concentration in finishing pigs.

Item	CTR	FJJG5	FJJG10	*p*-Value
Digestive enzyme activity in jejunum content				
Disaccharidases, U/mg	20.65 ± 4.38 ^b^	29.00 ± 4.91 ^a^	19.41 ± 4.04 ^b^	0.004
Lipase, U/g	27.74 ± 7.26 ^b^	59.08 ± 9.09 ^a^	30.83 ± 8.82 ^b^	<0.001
IgA concentration, μg/mgprot				
Jejunum	142.27 ± 12.76	135.24 ± 10.98	141.70 ± 14.48	0.585
Colon	119.55 ± 24.06	122.55 ± 9.74	128.24 ± 10.49	0.648

Duncan’s test was used for multiple comparisons of the mean. ^a,b^ Means in the same row without common superscripts differ significantly (*p* < 0.05).

**Table 8 animals-15-00459-t008:** Effects of fermented straw on intestinal gene expression in finishing pigs.

Item	CTR	FJJG5	FJJG10	*p*-Value
Jejunum				
*MUC2*	1.00 ± 0.35 ^b^	2.89 ± 0.83 ^a^	0.65 ± 0.18 ^b^	<0.001
*MMP13*	1.00 ± 0.26	0.94 ± 0.22	0.87 ± 0.30	0.692
*KCNJ13*	1.00 ± 0.26	0.87 ± 0.20	0.97 ± 0.28	0.638
*NHE3*	1.00 ± 0.21	1.19 ± 0.32	1.17 ± 0.48	0.607
*APOA1*	1.00 ± 0.28	0.65 ± 0.15	0.75 ± 0.44	0.175
*APOA4*	1.00 ± 0.18 ^b^	1.46 ± 0.34 ^a^	0.74 ± 0.27 ^b^	0.001
*SLC7A7*	1.00 ± 0.24 ^a^	0.63 ± 0.18 ^b^	0.72 ± 0.23 ^b^	0.026
*LPL*	1.00 ± 0.28 ^b^	1.52 ± 0.34 ^a^	1.23 ± 0.17 ^ab^	0.016
*IL10*	1.00 ± 0.15 ^a^	0.57 ± 0.17 ^b^	0.53 ± 0.18 ^b^	<0.001
*Colon*				
*MUC2*	1.00 ± 0.19	0.79 ± 0.25	0.87 ± 0.30	0.354
*MMP13*	1.00 ± 0.29 ^a^	0.81 ± 0.26 ^a^	0.49 ± 0.20 ^b^	0.011
*KCNJ13*	1.00 ± 0.26 ^a^	1.07 ± 0.32 ^a^	0.58 ± 0.17 ^b^	0.010
*NHE3*	1.00 ± 0.11	1.06 ± 0.21	1.14 ± 0.18	0.402
*APOA1*	1.00 ± 0.28	0.98 ± 0.38	0.65 ± 0.17	0.093
*APOA4*	1.00 ± 0.24 ^a^	0.69 ± 0.21 ^b^	0.33 ± 0.12 ^c^	<0.001
*SLC7A7*	1.00 ± 0.29 ^a^	0.81 ± 0.10 ^ab^	0.59 ± 0.13 ^b^	0.007
*LPL*	1.00 ± 0.14 ^a^	0.86 ± 0.08 ^ab^	0.73 ± 0.22 ^b^	0.034
*IL10*	1.00 ± 0.27 ^a^	0.73 ± 0.22 ^b^	0.65 ± 0.08 ^b^	0.024

MUC2: mucin 2; MMP13: matrix metalloproteinase-13; KCNJ13: potassium inwardly rectifying channel, subfamily J, member 13; NHE3: Na(+)-H(+) exchanger 3; APOA1: apolipoprotein A-I; APOA4: apolipoprotein A-IV; SLC7A7: solute carrier family 7 member 7; LPL: lipoprotein lipase; IL-1β: interleukin-10. Duncan’s test was used for multiple comparisons of the mean. ^a,b^ Means in the same row without common superscripts differ significantly (*p* < 0.05).

## Data Availability

If necessary, the first author can be contacted by email regarding the original data of the full text.

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
