# Peer review of "Dietary Effects of Different Proportions of Fermented Straw as a Corn Replacement on the Growth Performance and Intestinal Health of Finishing Pigs"

_animals, 2025, doi:10.3390/ani15030459_

Round 1

Reviewer 1 Report

Comments and Suggestions for Authors

Dear authors,

Many thanks for this piece of work which I read with great interest. Despite the experimental design is correctly developed I have more than one concern related to the idea behind the experiment and the information provided.The paper is not adequately contextualized and I fell that the missing information are somewhat functional to the scope. Moreover, I have concerns on the conflict of interest. 

I can accept the administration to ruminants but not to pigs in terms of basic physiology of nutrition ad feeding habits. I can't figure out the hypothesis in the background which cannot be aligned in your introduction. I feel that this is a major aspect that makes the whole experiment weak before my eyes, unless you provide real justification related to pigs and not to ruminants..

My concerns in fact are related to the economic convenience for the pork sector and sustainability of the technology behind the fermentation of poor feeds,which can be reasonably deployed to ruminants, but not to single stomach omnivores. In addition the effects of such inclusion in finishing pigs do not report valuable outcomes in clinical, sustainable and environmental aspects of such a practice, a priori. For straw is not a by-product but has implications both as feed and litter material, this might be of interest to increase the access to relative nutrients in ruminants, but not in pigs. And indeed, the inclusion of 10% replacing corn proves this (which is expected). In my opinion the idea goes to the opposite direction of sustainable and environmentally friendly feeding approach.

Author Response

Dear reviewer 1:

Thank you very much for reviewing our manuscript. We would like to thank you for your suggestions and comments. These suggestions are thoughtful and constructive, and help to improve the quality of this manuscript. Thanks for your suggestion, I will make the following point-to-point reply.

Comments1 ‘The paper is not adequately contextualized’, 

Response: Although raw straw is difficult to digest, the availability of straw nutrients can be improved by fermentation technology. This study is a preliminary attempt to use fermented straw in different proportions to partially replace corn raw materials in feed, at least to some extent. Therefore, we stated in the last sentence of the second paragraph that "To fully explore and utilize the potential value of crop straw by using biotechnology is one of the most effective means to alleviate the global shortage of food resources". The results showed that 5% instead of 10% fermented straw can improve lipid metabolism and colonic flora structure of fattening pigs, and has no adverse effects on the growth performance and intestinal health of fattening pigs. In addition, due to the low cost of straw, the cost of 5% fermented straw in this experiment is only about 1/2 of corn, after our cost accounting. This result has considerable economic significance for practical production. Developing new feed material to replace current raw feed materials such as corn is an important and arduous task in the current feed industry. Our experiments have been carried out in close cooperation with farms. Currently, low levels of fermented straw are used in fattening pig feed on farms, which saves some costs without reducing production performance. We believe that this has certain significance, especially for areas that rely on imported corn and have a large amount of waste straw.

Comments 2, ‘I have concerns on the conflict of interest.’ 

Response: the authors declare that there is no conflict of interest. Even though we have cooperation with farms and firms, we have the ultimate intellectual property.

Comments 3, ‘I can accept the administration to ruminants but not to pigs’

Response:'improving microbiota structure is one of the important goals in livestock and the poultry industry', this sentence not only refers to ruminants, but also all animals. More and more studies have shown the application of fermented straw in pigs (In text, 13-14), which means the feasibility of fermented straw on monogastric animals. Although there are not many studies, our mission is to explore more possibilities. This is also the significance of trying to use fermented straw, and the results will provide reference value for more people.

Comments 4, ‘economic convenience for the pork sector and sustainability of the technology’

Response: we have already answered it in question 1. In short, fermented straw replaces 5% of corn raw materials, which reduces production costs without reducing growth performance. This is undoubtedly an advantage. In addition, the ‘CO2-neutralization procedure’ fermentation method is easy to operate in production and is convenient for large-scale promotion. Therefore, in terms of economy and sustainability, the application prospects of fermented straw may be relatively broad.

Thanks again for your suggestion.

Reviewer 2 Report

Comments and Suggestions for Authors

Comments to the Authors of manuscript number animals-3320311 entitled “Effects of different proportions of fermented straw replacing corn in the diet on growth performance and intestinal health in finishing pigs”

1. The word "Employees" is likely a typographical error and should be corrected to "Employing."

2. 72-73 - The sentence structure is awkward. The intended meaning is obscured - "widely used" and "still in its infancy."

3. 46-47-  Typographical or logical error ("46 decades").

4. 72-73: Awkward sentence structure and poor punctuation.

5. 93- small letter

6. 99 – how frequent was that record?

7. why the experiment lasts 39 days? It should be explained, the basal reason

8. 130- what segment?

Author Response

Dear reviewer 2:

Thank you very much for reviewing our manuscript. We would like to thank you for your suggestions and comments. These suggestions are thoughtful and constructive, and help to improve the quality of this manuscript. Thanks for your suggestion, I will make the following point-to-point reply.

Comments 1: The word "Employees" is likely a typographical error and should be corrected to "Employing.".

Response 1: Thanks for your comments. We have checked and revised the word in line 74.

Comments 2: 72-73 - The sentence structure is awkward. The intended meaning is obscured - "widely used" and "still in its infancy.".

Response 2: we have revised the sentence into ‘Currently, corn straw fermented feed has been widely used as in ruminants, but is rarely applied in pigs and other monogastric animals’in line 73.

Comments 3: 46-47- Typographical or logical error ("46 decades").

Response 3: Thanks for your comments. Yes, it was a line number.

Comments 4: 72-73- Awkward sentence structure and poor punctuation.

Response 4: we have revised the sentence into ‘Currently, corn straw fermented feed has been widely used as in ruminants, but is rarely applied in pigs and other monogastric animals’in line 73.

Comments 5: 93- small letter.

Response 5: Thanks for your comments. We have modified this description in the text.

Comments 6: 99- how frequent was that record?

Response 6: Body weight and food intake were recordedevery two weeks (D0, D14, D28) until the pigs from the control group reached to commercial finished weight (130kg)(on D39).We have added the sentence in text (line 100).

Comments 7: why the experiment lasts 39 days? It should be explained, the basal reason.

Response 7: Body weight and food intake were recordedevery two weeks (D0, D14, D28) until the pigs from the control group reached to commercial finished weight (130kg)(on D39).We have added the sentence in text (line 100).

Comments 8: 130- what segment? 

Response 8: Jejunum and Colon.

Thanks again for your help to our manuscript.

Round 2

Reviewer 1 Report

Comments and Suggestions for Authors

Dear Authors,

Many thanks for your replies to my critical questions, that did not, however, come along with substantial modifications to the manuscript. For your activities have several flaws that were not correctly addressed in the text of your 'revised' version, I must consider your revision unsatisfactory. 

Author Response

Dear Reviewer,

Thanks for your comments. As we explained earlier,  This result has considerable economic significance for practical production. Developing new feed material to replace current raw feed materials such as corn is an important and arduous task in the current feed industry. Our experiments have been carried out in close cooperation with farms. Currently, low levels of fermented straw are used in fattening pig feed on farms, which saves some costs without reducing production performance. We believe that this has certain significance, especially for areas that rely on imported corn and have a large amount of waste straw. Our research not not obstacles by the traditional view that the fermented corn straw was not suitable for monogastric animals. We will continue our study and contribute more findings to academic research.

Best,

All authors